# Screening for Multifarious Plant Growth Promoting and Biocontrol Attributes in *Bacillus* Strains Isolated from Indo Gangetic Soil for Enhancing Growth of Rice Crops

**DOI:** 10.3390/microorganisms11041085

**Published:** 2023-04-21

**Authors:** Shikha Devi, Shivesh Sharma, Ashish Tiwari, Arvind Kumar Bhatt, Nand Kumar Singh, Monika Singh, Ajay Kumar

**Affiliations:** 1Department of Microbiology, Himachal Pradesh University, Summerhill, Shimla 171005, India; shikhasharma003@gmail.com; 2Department of Biotechnology, Motilal Nehru National Institute of Technology Allahabad, Prayagraj 211004, India; ashishtiwari13@gmail.com (A.T.); nksingh@mnnit.ac.in (N.K.S.); 3Department of Biotechnology, Himachal Pradesh University, Summerhill, Shimla 171005, India; bhtarvind@yahoo.com; 4Department of Biotechnology, School of Applied and Life Sciences, Uttaranchal University, Dehradun 248007, India; monikasingh_bhu@yahoo.com; 5Department of Zoology, Pachhunga University College Campus, Mizoram University (A Central University), Aizawl 706001, India; kaushalpuc@gmail.com; 6Centre of Advanced Study in Botany, Banaras Hindu University, Varanasi 221005, India

**Keywords:** inoculants, indo gangetic plains, 16S rDNA gene, hydrogen cyanide (HCN), indole-3-acetic acid (IAA)

## Abstract

Multifarious plant growth-promoting *Bacillus* strains recovered from rhizospheric soils of the Indo Gangetic plains (IGPs) were identified as *Bacillus licheniformis* MNNITSR2 and *Bacillus velezensis* MNNITSR18 based on their biochemical characteristics and 16S rDNA gene analysis. Both strains exhibited the ability to produce IAA, siderophores, ammonia, lytic enzymes, HCN production, and phosphate solubilization capability and strongly inhibited the growth of phytopathogens such as *Rhizoctonia solani* and *Fusariun oxysporum* in vitro. In addition, these strains are also able to grow at a high temperature of 50 °C and tolerate up to 10–15% NaCl and 25% PEG 6000. The results of the pot experiment showed that individual seed inoculation and the coinoculation of multifarious plant growth promoting (PGP) *Bacillus* strains (SR2 and SR18) in rice fields significantly enhanced plant height, root length volume, tiller numbers, dry weight, and yield compared to the untreated control. This indicates that these strains are potential candidates for use as PGP inoculants/biofertilizers to increase rice productivity under field conditions for IGPs in Uttar Pradesh, India.

## 1. Introduction

Nutrient depletion, the inappropriate use of chemical fertilizers and the loss of biodiversity are major threats involved in the decline of crop production, productivity, and soil health. In this context, the shift from chemicals to soil bacteria or plant growth-promoting rhizobacteria (PGPR) is a non-hazardous sustainable approach to increasing agricultural productivity, replenishing soil nutrients, and controlling phytopathogens in an eco-compatible manner. Keeping these facts in mind, the rhizosphere of rice plants associated with IGPs was explored for the isolation of multifarious PGP *Bacilli* and showed plant growth promoting and biocontrol activities under *vitro* and *in vivo* conditions.

PGPR is a diversified group of beneficial soil bacteria that live in the rhizosphere region of the plant and are capable of stimulating and enhancing plant growth or yield when cultivated in association with a host plant. Plant growth-promoting rhizobacteria use one or more types of mechanisms, i.e., direct and indirect, to enhance host plant growth. The solubilization of phosphate and production of ammonia in the root-soil environment directly promote plant growth by acting as macronutrients, while the synthesis of phytohormones, i.e., indole-3-acetic acid (IAA) generally accelerates root growth at low concentrations. Siderophores that chelate iron, such as HCN and other lytic enzymes viz., protease, amylase, cellulase, lipase, and ACC (1-aminocyclopropane-1-carboxylate) deaminase, etc., are produced in the rhizosphere region by PGP microorganisms (including bacteria and mycorrhizal fungi) and can indirectly support plant growth by suppressing the deleterious effects of biotic and abiotic stresses [1,2]. (Goswami Over the past few years, a growing number of studies have been conducted on PGPR identification, mainly because the role of the rhizosphere as an ecosystem has gained importance in the functioning of the biosphere. Various species of plant-associated rhizobacteria, including species of *Pseudomonas, Bacillus, Azospirillum, Azotobacter, Enterobacter, Alcaligenes, Arthrobacter, Burkholderia, Serratia,* and *Klebsiella,* have been reported to enhance plant growth [3,4]. Among the myriads of rhizobacteria that thrive in the rhizosphere of plants, Gram-positive spore-forming bacilli have attracted special attention because of their ability to resist pesticides and survive under harsh environmental conditions such as high salinity, extreme pH, and high temperatures in the field [5], providing more consistent results under natural conditions [6].

Rice (*Oryza sativa*) is the most widely grown cereal crop grown in the different locations of IGP regions and is an important staple food for more than half of the world’s population. Studies by various researchers have shown that fertilizers are an essential component of modern agriculture as they provide important plant nutrients. However, the indiscriminate and excessive use of chemical fertilizers and pesticides can lead to unforeseen health hazards and environmental impacts [7]. Considering these facts, it is imperative to find alternative strategies that ensure competitive yields while protecting soil health [8]. Soil microbes and PGPR play an essential role in the agriculture and food sectors due to their pivotal functions in recycling plant nutrients, symbiotic relationships, the detoxification of harmful chemicals, and the overall promotion of plant growth without disturbing the soil ecosystem. Alterations in rhizosphere microbial communities and soil microbial activity are closely related to soil fertility, environmental quality, and agro-climatic conditions [9]. Previous reports have suggested that rhizobacterial isolates that are best adapted to one environment may not perform in the same way to evoke plant response in others due to the conditions prevailing in a new environment [8]. The survival of PGP inoculants in the rhizosphere is affected by a wide range of abiotic stress conditions, viz., flooding stress, high salinity, extreme pH, and high temperature [3,10]. Moreover, there is very limited information on the promotion of plant growth in rice (*Oryza sativa*) through the application of multifarious PGPR in agricultural soils of the Indo-Gangetic Plain, India. The screening of *Bacillus* strains in this study could provide multipurpose bioinoculants for agricultural crops that have PGP and biocontrol potential coupled with stress tolerance activity. At present, the research focus is moving toward helping poor farming communities. especially those living in the rural areas of Uttar Pradesh in the IGP region, whose income and livelihood depend solely on agriculture. Hence, the aim of the present study was to isolate and characterize bacteria of the genus *Bacillus* from the rhizosphere of *Oryza sativa* that exhibit PGP and biocontrol activities *in vitro* or *in vivo* so that they can be used as potential biofertilizers for the sustainable productivity of rice.

## 2. Material and Methods

### 2.1. Collection of Soil Samples from the Paddy Rhizosphere

The present investigation was undertaken with the aim of selecting multifarious plant growth-promoting *Bacillus* strains from the rice rhizosphere. A detailed survey of paddy growing areas was carried out to systematically evaluate the cropping pattern of the IGP region, and Bijarkala, Sidhvan, and Barkaccha were selected for the collection of soil samples. The study site was located in the eastern region of the mid-Gangetic plain of Uttar Pradesh, India, which lies between a 25°10′ N latitude to 82°44′ E longitude. Sampling was carried out during the months of September and October 2012. Rhizospheric soil samples were collected and placed in plastic bags and stored at 4 °C in the Laboratory of the Department of Biotechnology, MNNIT Allahabad, for further analysis.

#### Isolation and Maintenance of Bacterial Isolates

Bacteria were isolated from the rhizospheric soil samples by using serial dilution agar plate technique using 1 g of soil, and CFU (colony forming units) were recorded after 24–48 h of incubation at 28 ± 1 °C [11].

Initially, a number of bacteria were isolated and routine observations revealed that various isolates were common, which were then excluded. Finally, 8 isolates were selected and analyzed for their morphological and staining features and maintained on nutrient agar (Himedia) slants at 4 °C.

### 2.2. Characterization of Soil Bacterial Isolates for Various PGP Attributes

#### 2.2.1. Phosphate Solubilization

Gram-positive rod-shaped bacterial isolates were screened for tri-calcium phosphate (TCP) solubilization as per the methodology described by Pereira et al. [12]. On Pikovskaya agar (Himedia) containing TCP as the only phosphate source (pH 7.0 ± 0.2), a loop full of each culture was spot inoculated or streaked on the surface of the agar plates and incubated at 30 ± 0.1 °C for 5 days. A clear halo around the bacterial colony indicated the phosphate-solubilizing activity of the test isolates [13].
Phosphate solubilization index (PSI) = colony diameter + halozone diameter/colony diameter

#### 2.2.2. IAA Production and Siderophore Production

IAA production was estimated with the Salkowski reagent using a colorimetric assay with a nutrient broth containing 0.1% DL tryptophan (Sigma-Aldrich) [14]. The production of siderophores by all isolates was detected on blue agar plates with chrome azurol sulphonate (CAS) (Sigma-Aldrich) dye as per the methodology developed by Schwyan and Neilands (1987). A loop full of 24 h old bacterial cultures was spot inoculated onto the center of the CAS agar plates and incubated at 30 ± 0.1 °C for 9–17 days.

#### 2.2.3. HCN Production

The screening of all isolates for HCN production was performed according to the method of Castric on nutrient agar medium amended with 4.4 g/L glycine using Whatman filter paper No. 1 soaked in 0.5% picric acid solution (Merck) [15].

#### 2.2.4. Ammonia Production

Bacterial isolates were checked for ammonia production using the method of Cappuccino and Sherman [16]. The 12 h old bacterial cultures were inoculated in 10 mL of peptone broth and incubated for 48 h at 28 ± 0.1 °C. The appearance of a faint yellow to dark brown color upon the addition of a 0.5 mL Nessler reagent (Merck) to the cultures indicated the production of ammonia.

#### 2.2.5. ACC Deaminase Activity

ACC deaminase activity was detected by growing the bacterial isolates on agar plates containing Dworkin and Foster (DF) salt minimal medium with 3 mM ACC (Sigma-aldrich) as the sole source of nitrogen [17]. Plates streaked with the bacterial cultures were incubated at 28 ± 0.1 °C and examined for bacterial growth after 48 h.

### 2.3. In Vitro Screening of the Bacterial Isolates for their Antagonistic Activities against Soil Borne Plant Pathogens

#### 2.3.1. Fungal Culture

Plant pathogens *Rhizoctonia Solani* and *Fusarium oxysporum* were obtained from the Microbial type culture collection (MTCC) Chandigarh, India, and were maintained on potato dextrose agar (PDA) (Himedia) slants at 4 °C for further use.

#### 2.3.2. Qualitative Evaluation of Antagonism Due to Diffusible Compounds

To investigate antagonism due to diffusible compounds, a lawn of the test fungus (*R. solani* and *F. oxysporum)* was grown on potato dextrose agar (PDA) plates [18]. The *in vitro* mycelial growth inhibition of the test fungus was studied using a dual culture technique [19]. Discs 7 mm in diameter were cut from the fungal lawns with the help of a borer and placed on another plate (containing nutrient agar and PDA in a 1:1 ratio). The bacterial culture was inoculated approximately 2 cm away from the fungal disc. The percent growth inhibition (PGI) was calculated using the following formula:PGI = [(R1 − R2)/R1 × 100]
where R1 is the maximum radius of the fungal growth away from the bacterial colony, and R2 is the radius of fungal growth opposite the bacterial colony.

### 2.4. Production of Cell Wall Hydrolytic and Industrially Important Enzymes by Potential Bacillus Strains

#### 2.4.1. Chitinase and Protease Production

Protease and chitinolytic activities were determined by plating the bacteria on skimmed milk agar (Himedia) and chitin (Sigma-Aldrich) agar plates, as described by Naik et al. [20]. A zone of clearance around the colony after 5 days of incubation at 30 °C showed a positive result for protease and chitinase production.

#### 2.4.2. Pectinase Activity

Pectin degrading enzymes were assayed using an M9 medium amended with 4 g of Pectin (Sigma-Aldrich) per liter. Plates were incubated at 28 °C for 2 days. The appearance of a clear halo around the colonies indicated pectinase production [20].

#### 2.4.3. Cellulase, Lipase and Amylase Activity

Cellulolytic and amylase activity was determined by plating the bacteria on carboxy methyl cellulose (CMC) (Merck) agar and starch casein agar (Himedia) plates. A clear zone around the colony indicated a positive result for cellulase and amylase production. The bacterial isolates were also screened for lipolytic activity by plating the bacteria on a nutrient agar medium amended with olive oil (Himedia) and rhodamine (Merck) [21]. The development of an illuminating zone around the test isolate was considered positive when examined under UV light.

### 2.5. Morphological and Biochemical Characterization

Morphological and staining features were examined by microscopic observation. Various biochemical tests, including indole, methyl red, Voges-Proskauer and citrate (IMViC), oxidase, catalase, starch and gelatin hydrolysis, motility, nitrate, and urease reduction, etc., were performed as per the methodology of Cappuccino and Sherman [16].

### 2.6. Molecular Characterization of the Potential Isolates

Genomic DNA was isolated using the Ultra Pure Genomic DNA spin minipreps kit from bacteria (Medox) and using the manufacturer’s instructions. DNA samples were subjected to gel electrophoresis in 0.8% agarose and were visualized by illumination with ≈300 nm UV light. 16S rDNA was amplified using the universal eubacterial primers fD1 (5′-CCGAATTCGTCGACAACAGAGTTTGATCCTGGCT AG-3′) and rp2 (5′-CCCGGGATCCAAGCTTACGGCTACCTTGTTACGACTT-3′) [22] and using the thermocycler (Bio-Rad, S1000). The amplified DNA was visualized in 1% agarose, and the PCR product was eluted using the Bioserve Gel Elution Kit (catalog No 2021). Internal sequencing primers were used for sequencing the ~1.5 kb region. Sequencing was performed using four different primers, viz., 16SEQ2R, 16SEQ3F, INS16SREV, and 16SEQ4R, that were designed in the conserved regions of 16SrDNA. The identity of the isolates was determined by a BLAST search. The Nucleotide Basic Local Alignment Search Tool (BLASTn) was performed to search for the similarity of isolated 16S rRNA gene sequences with other sequences deposited in GenBank [23]. Phylogenetic and molecular evolutionary analysis was conducted using Seaview version 5 software. The phylogenetic tree was constructed using the maximum parsimony method with 100 bootstrap values [24].

### 2.7. Abiotic Stress Tolerance Activity of the Potential Bacillus Strains

Potential *Bacillus* strains (SR2 and SR18) were checked for their ability to tolerate different abiotic stresses, viz., high temperature, salinity, and drought using a nutrient broth (NB). The growth of all isolates was recorded, using a spectrophotometer (Eppendorf) at 600 nm, with the uninoculated medium serving as a blank. Test isolates were considered stress tolerant when an optical density (OD) of 0.1 was measured [3].

#### 2.7.1. Drought Resistance

The bacterial isolates were tested *in vitro* for their tolerance to water stress according to the methodology of Sandhya et al. [25]. A nutrient broth medium with different water potentials (−0.05, −0.15, −0.30, −0.49, −0.73 Mpa, and −1.2 Mpa) was prepared by adding the proper concentrations of polyethylene glycol (PEG 6000) (Merck). The broth was then inoculated with 1% of the bacterial culture overnight in the nutrient broth, and incubated at 28 ± 0.1 °C for 24 h, after which OD was recorded at 600 nm. 

#### 2.7.2. Temperature Resistance

The bacterial isolates were tested for temperature resistance by incubating them at different temperatures ranging from 10 °C to 50 °C for 24 h [3]. The bacterial isolates were inoculated into nutrient broth and the inoculated broth was then incubated at the appropriate temperature for 24 h, after which OD was recorded at 600 nm.

#### 2.7.3. Salt Tolerance

Bacterial isolates were tested for salt tolerance by inoculating the bacterial culture on nutrient broth amended with a salt concentration ranging from 0.5% to 20% [3].

### 2.8. In Vivo Experiment to Evaluate the Efficacy of Potential Isolates to Promot Plant Growth under Green House Conditions

#### 2.8.1. Pot Culture Assay under Greenhouse Conditions

To determine the PGP potential of rhizobacteria, rice seeds (variety Samba Mahsuri) were treated with four treatments with three replicates and were observed for various growth parameters in the greenhouse of the Department of Biotechnology, MNNIT Allahabad. The treatments were as follows (i) A: SR2 (*Bacillus licheniformis*) (single inoculation) (ii) B: SR18 (*Bacillus velezensis*) (single inoculation) (iii) combination of A + B (SR2 + SR18) and (iv) a control (untreated).

For the experimental setup, plastic pots were first sterilized with 0.7% sodium hypochlorite or 70% ethanol and filled with autoclaved sterile soil before the rice seeds were sown. The seeds of the rice plants were surface disinfected by washing in 70% ethanol for 1 min, followed by washing with a 4% solution of the sodium hypochlorite solution for 5 min and rinsing with sterile distilled water for five to seven times [26]. Seven seeds of each treatment were sown under the 2 cm soil surface in each plastic pot, and after thinning, three seeds per pot were retained, including the control treatment.

The greenhouse experiment was conducted with a photoperiod cycle of 14 h of light and 10 h of darkness at 25 °C to 30 °C and a humidity of about 80%. Further, the experiment was completed within 150 days, i.e., from seed inoculation to rice grain harvest. The plants were irrigated three times a day through an automatic and computer-controlled drip irrigation system. During the course of the experiment, various parameters, viz., shoot length, root length, dry weight, and yield were studied. Root length was measured using a meter scale. To determine root length, three individual plants were randomly selected from each experimental pot and measured with a meter scale after uprooting the selected plants. The dry matter content of plant seedlings was determined by drying a sample in an oven, usually at 60° C, until a constant weight was reached.

#### 2.8.2. Seed Bacterization

Rice seeds were bacterized/bacterially treated according to the method of Weller and Cook [27]. Liquid cultures of selected bacterial strains were grown in a nutrient broth at 120 rpm at 28 °C for 48 h and centrifuged at 7100 rpm at 4 °C for 15 min to obtain the pellet. The pellets were washed with sterile distilled water and resuspended in distilled water to obtain a cfu in the range of 10^8^ CFU/ mL. In the case of using a mixture of the two PGP strains, an equal volume of both strains was mixed instantaneously before use. This suspension was then mixed with 1% CMC to form a slurry which was coated to the surface of the rice seeds.

### 2.9. Statistical Analysis

All experiments were performed with three replicates and subjected to an analysis of variance (ANOVA) using *RAUSTAT Windows V-1* 2003. Significant differences between means were separated according to Duncan’s multiple range test.

## 3. Results and Discussion

To isolate pure cultures of bacteria associated with the rhizosphere of rice, the serial dilution technique was used, and a number of bacteria were isolated on nutrient agar (NA) and maintained on agar slants at 4 °C. Routine observations revealed that various isolates were common and excluded. Finally, eight isolates were selected and analyzed for their morphological and staining features. Gram-positive rods (designated SR2, SR3, SR11, SR13, SR18, SR20, SR56, and SR58) obtained in the present study were subjected to various PGP attributes *in vitro* and *in vivo* due to their sporulating ability, which was superior to other non-spore formers and facilitated their survival under extreme conditions viz., high temperatures, extreme pH, and environmental stresses, and they were also ideal for use as commercial bioinoculant formulations [5,28,29].

### 3.1. Plant Growth Promoting Activity of Bacillus Isolates Recovered from Soils of the IGP Region

#### 3.1.1. IAA Production

The principal auxin found in higher plants was IAA, which regulates various physiological processes, including root initiation, cell enlargement, and the stimulation of cell division, and results in an increase in the root surface area while enabling the plant to take up significantly more nutrients from the soil [30]. Ample studies have shown that 80% of soil microbial communities associated with the rhizosphere of various crops, i.e., wheat, corn, and sugarcane have the ability to produce plant hormones such as IAA. In our study, all candidate isolates were capable of producing IAA in the presence of tryptophan. A comparatively high level of IAA was found in the bacterial isolate of SR2 (90.64 µg/mL) followed by SR18 (87.15 µg/mL) compared to the other isolates (Table 1).

#### 3.1.2. Phosphate Solubilization

Phosphorus is the second most abundant plant nutrient available in the soil after nitrogen and is classified as a major plant nutritive element. A large proportion of soluble inorganic phosphate added to the soil as a chemical fertilizer was fixed in an insoluble form soon after application and was no longer available to the plants [31,32]. Plants utilize only inorganic phosphorus, and organic phosphorus compounds must first be hydrolyzed by phosphatase enzymes, which mostly originate from plant roots, through the action of bacteria [33]. In the present study, seven isolates had the ability to solubilize phosphorus, namely SR3, SR2, SR11, SR13, SR20, SR56, and SR18 (Table 1). The isolate SR2 showed the highest phosphate solubilization zone on Pikovskaya agar with a phosphate solubilization index of 11 mm compared to the other bacterial isolates.

#### 3.1.3. Siderophore Production

Siderophore production is an important trait of soil bacteria that chelates iron in the rhizosphere and prevents fungal pathogens from flourishing in iron-limiting conditions [34,35]. Siderophores are low molecular-weight iron-complexing compounds that are secreted by microbes in response to iron limitations to take up iron from insoluble forms through mineralization and sequestration [4,36,37]. Of the isolates studied, all are capable of producing orange halos around colonies, indicating positive siderophore production in CAS agar plates. Amongst all of these, isolate SR18 displayed maximum siderophore production compared to the others.

### 3.2. In Vitro Antifungal Activity of the Potential Bacillus Isolates

In the present study, the *in vitro* antagonistic effect of potential candidates against soil-borne pathogens was observed using the dual plate culture technique [4,18,19,38]. The previous literature documented how various PGPR, viz., *Penibacillus, Microbacterium, Bacillus,* and *Klebsiella* isolated from Korean rice cultivars exhibited a significant zone of inhibition against soil-borne fungi, *F. oxysporum,* and *R. solani*. In this study, all isolates were evaluated for their biocontrol activity against test pathogens under *in vitro* conditions. Out of the screened isolates, four isolates, SR13, SR2, SR18, and SR56, were able to inhibit the mycelial growth of *F. oxysporum* and *R. solani* (Table 1). SR2 and SR18 showed a significant zone of inhibition for the growth of *Fusarium oxysporum*. The maximum growth inhibition exhibited by SR2 and SR18 against *Fusarium oxysporum* was 67% and 48%, respectively, compared to the control. SR2 and SR18 isolates also showed a significant reduction in mycelial growth by 62% and 50%, respectively, against *R. solani*. Our results are in agreement with the findings of Ahmad et al. [39], who reported that *B. subtilis* isolate B4 showed the highest inhibition percentage (i.e., 47.1%) against *R. solani*, followed by isolate B7 which achieved 44.9%. Numerous studies have demonstrated that PGPR produced a variety of compounds, viz., antibiotics, siderophores, HCN, metabolites, lytic enzymes, and others, which can act as effective biocontrol agents against a wide range of pathogens [4,5]. *Bacillus* and *Pseudomonas* are the most extensively studied plant-associated genera known for their promising biocontrol activities [4,38]. In addition, studies by Kumar et al. [3] reported that fluorescent pseudomonads and certain *Bacillus* species had a greater antagonistic ability against the bacterial and fungal root diseases of crops. *P. fluorescens* isolates also displayed their antagonistic potential to inhibit important fungal pathogens, viz., *R. solani* and *Pyricularia grisea,* in a dual culture plate assay [20]. The biological control of phytopathogens can occur through a diverse array of mechanisms, usually referred to as suppression, direct parasitism, antibiosis, competition, predation, hypovirulence, and induced systemic resistance [1,4]. Therefore, isolates with biocontrol activity play a beneficial role in plant growth and development and are considered an important component of management practices to achieve greater crop yields on a sustained basis.

#### 3.2.1. HCN Production

Ample studies have demonstrated that cyanide production by PGPR affects the respiratory system of plant pathogenic fungi and leads to the inhibition of their mycelial growth [15,26]. HCN production is observed less frequently compared to other PGP traits. The current investigation revealed that only two isolates, SR3 and SR18, were capable of HCN production (Table 1).

#### 3.2.2. Ammonia Production

The production of ammonia by PGPR is another important trait that directly promotes plant growth and productivity [40]. In the present study, six isolates have the ability to produce ammonia in peptone water (Table 1).

#### 3.2.3. ACC Deaminase Activity

Several studies have shown that ACC, a key precursor of plant ethylene, has beneficial effects on the growth of plants by reducing the concentration of ACC under a wide range of different abiotic stresses, viz., salt stress, flooding stress, heavy metal stress, and pathogen attack [41,42]. Out of the screened isolates, five were capable of producing ACC as the sole source of nitrogen, and the result of ACC deaminase activity is given in Table 1.

### 3.3. Screening of Industrially Important and Cell Wall Hydrolytic Enzymes by Potential Bacillus Isolates

The results from various researchers demonstrate that the ability to produce siderophores, HCN, and other lytic enzymes, viz., chitinase, protease, pectinase, and cellulase by PGPR in the rhizospheric can indirectly enhance the growth of plants by suppressing the deleterious effects of biotic stress [1]. None of these isolates were positive for the production of chitinase and pectinase, whereas the isolates were able to produce siderophore and various other enzymes, i.e., cellulase, protease, lipase, and amylase (Table 1). Only three isolates, SR3, SR2, and SR56, produced the enzyme cellulase, while the rhizobacterial isolates SR13, SR2, SR18, and SR58 produced protease in the skim milk agar. The bacterial isolates SR13, SR2, and SR18 were found to be positive for lipase and amylase production compared to the other isolates.

### 3.4. Molecular Characterization of the Potential Isolates

Bacterial isolates SR2 and SR18 exhibited multifarious PGP characteristics and were identified as *Bacillus licheniformis* MNNIT SR2 and *Bacillus velezensis* MNNIT SR18, respectively, based on morphological and biochemical characteristics (Table 2) and 16 s rDNA gene sequence analysis. The phylogenetic tree (constructed using the maximum-parsimony method) has a percent similarity and genebank accession no. of multifarious *Bacillus* strains, which are presented in Figure 1 and Table 3.

### 3.5. Abiotic Stress Tolerance Activity of Potential Bacillus Strains Recovered from Soils of IGP Region

Studies by various researchers indicate that the survival of an introduced strain in a particular root soil environment is affected by a wide range of environmental stresses, such as flooding, extreme pH, high salinity, and high temperature [3,10]. Potential isolates SR2 and SR18 were evaluated in the present study for their ability to tolerate a wide range of extreme conditions. Both *Bacillus* strains in the present study were able to grow at high temperatures of 50 °C (Figure 2A). This could be due to their ability to form endospores, which facilitates their survival under harsh environmental conditions [5]. Drought is a major environmental problem and the most important abiotic stress that can cause huge productivity losses in arid and semi-arid regions where agriculture is completely dependent on rainfall. Recently, beneficial microorganisms/rhizobacteria have gained importance in stress management as they represent a promising eco-friendly approach to mitigate the adverse effects of stress and improve plant growth under such conditions [1]. In the present work, both strains also showed salt and drought tolerance and were found to be resistant to 10–15% NaCl and 25% PEG 6000, respectively (Figure 2B,C). In addition, strain *Bacillus velezensis* SR18 grew well under high salt concentrations (i.e., 5%, 8%, and 10%), as shown in Figure 2B, compared to 0.5% to 2%. Our result supports the findings of various researchers who have shown that *Bacillus velezensis* tolerates high salt concentrations of up to 11% NaCl [31]. To combat biotic stresses, viz., salinity, plants activate various physiological responses such as the synthesis of phytohormones, the production of antioxidants, and the regulation of nutrient uptake. Moreover, plant-associated bacteria could significantly promote plant growth under such harsh conditions by releasing growth-promoting substances and regulators [31].

### 3.6. Effect of Rice Associated Bacilli on Growth of Rice Plants Growing under Green House Conditions

The seed treatment of rice with the bacterial suspension *Bacillus licheniformis* strain MNNIT SR2 showed a significant increase in fresh shoot and root length (14.7% and 30.8%, respectively) and the dry weight of the shoot and root (53% and 35.6%, respectively) of rice plants after eight weeks compared to the untreated control in the pot culture assay (Table 4). Similar results were obtained with *Bacillus velezensis* MNNIT SR18 which also increased the fresh shoot and root length (17% and 20%) (Table 4) and the dry weight of the shoot and root (88.3% and 83.7%) (Table 4) of rice plants after eight weeks compared to the untreated control in the pot culture assay. When comparing the individual seed inoculations, the plants inoculated with the SR18 strain showed a significant increase in all plant growth parameters compared to the non-inoculated control plants and the plants treated with the SR2 strain (Table 4).In addition, the co-inoculation of *Bacillus* strains (SR2 and SR18) also stimulated the growth of rice plants under pot culture conditions and it was observed that fresh shoot and root length were significantly increased by 27% and 52.3%, and the dry weight of the shoot and root were increased by 90% and 91.7%, respectively (Table 4) compared with individual seed inoculation and the control. A significant increase in root number, leaf number, and tiller number was also observed in the treated rice plants compared to the untreated control. The results of these parameters with statistical analysis are presented in Table 4. Moreover, the results regarding grain yield showed a maximum increase in the yield (29.17 g/plant) in the case of the coinoculation of *Bacillus* strains, with a rapid increase of 15.45% over the control treatment. The promising results of the pot culture assay indicated that the combined inoculation of rice plant seeds with *Bacillus licheniformis* strain MNNIT SR2 and the *Bacillus velezensis* strain MNNIT SR18 showed a significant increase in overall plant growth promotion such as plant height, root length volume, tiller numbers, and dry weight compared to the individual seed treatments and untreated control (Figure 3A–C and Table 4). Similarly, Jabborova et al. [43] reported a significant enhancement in the growth of soybean plants due to the synergistic effect of the co-inoculation of *Bradyrhizobium japonicum* USDA110 and *Pseudomonas putida* NUU8 compared to inoculation with a single strain and the control.

Several species of the genus Bacillus, such as *B. amyloliquefaciens*, *B. aryabhattai*, *B. circulans*, *B. coagulans*, *B. licheniformis*, *B. megaterium*, *B. subtilis*, *B. thuringiensis*, and *B. velezensis* have previously been identified and characterized as PGPR and biocontrol agents [18]. They promote plant growth through various direct and indirect mechanisms, including nitrogen fixation, phosphate and potassium solubilization, phytohormone production, siderophores production, biosynthesis of antimicrobial and hydrolytic enzymes, the stimulation of induced systemic resistance (ISR), and antioxidant defense system in plants [3,18].

Ji et al. [13] reported that bacterial suspensions of endophytic diazotrophic bacteria SW521-L21, KW7-S22, KW7-S06, HS-R01, and CB-R05 showed a significant increase in the number of surviving plants in soil without chemical fertilizer amendments [13]. Likewise, Kapri and Tewari [44] and Chabot et al. [45] also reported an increase in shoot/root length and the dry matter of chickpea and maize plants, respectively, by inoculation with phosphate solubilizing PGP 95 strains. By contrast, Cardinale et al. [46] suggested in their study that screening based on pure culture assays might not be appropriate for recognition as a best plant growth promotion candidate and suggested that currently commonly used PGP screening strategies may need to be re-evaluated to detect promising plant growth promoters, which may increase the efficiency of PGPR investigations. In the present study, the results of pot trials revealed that these efficient treatments may be beneficial in improving rice crop potential in the vicinity of middle Gangetic plains.

In the present study, the individual inoculation of seeds and coinoculation of *Bacillus* strains (*Bacillus licheniformis MNNITSR2* and *Bacillus velezensis MNNITSR18*) showed a significant increase in overall plant growth, such as plant height, root length volume, tiller numbers, dry weight, and yield of rice plants in the pot culture assay without any chemical fertilizer amendments such as the murate of potash, diammonium phosphate and urea, etc., and can be considered as the best alternative with which to replace synthetic fertilizers, pesticides, and supplements in an eco-compatible manner. Overall, the *results* suggest that the presence of effective *Bacillus* strains exhibiting multiple PGP traits coupled with stress tolerance may be exploited as a potential bioinoculant agent for rice cultivation and sustainable crop management under field conditions in the IGPs of Uttar Pradesh (UP), India.

## Figures and Tables

**Figure 1 microorganisms-11-01085-f001:**
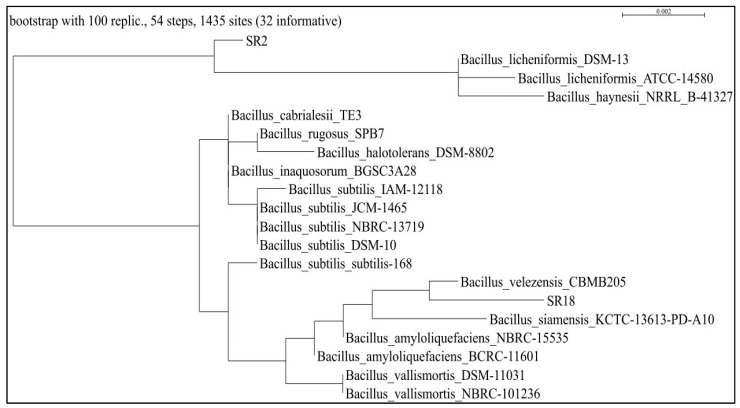
Phylogenetic tree analysis of multifarious PGPRs, viz., *SR2 Bacillus licheniformis* and *SR18 Bacillus velezensis*. This was conducted on the basis of 16SrRNA gene sequence alignment obtained from the rhizosphere of rice plants grown in the vicinity of IGP regions.

**Figure 2 microorganisms-11-01085-f002:**
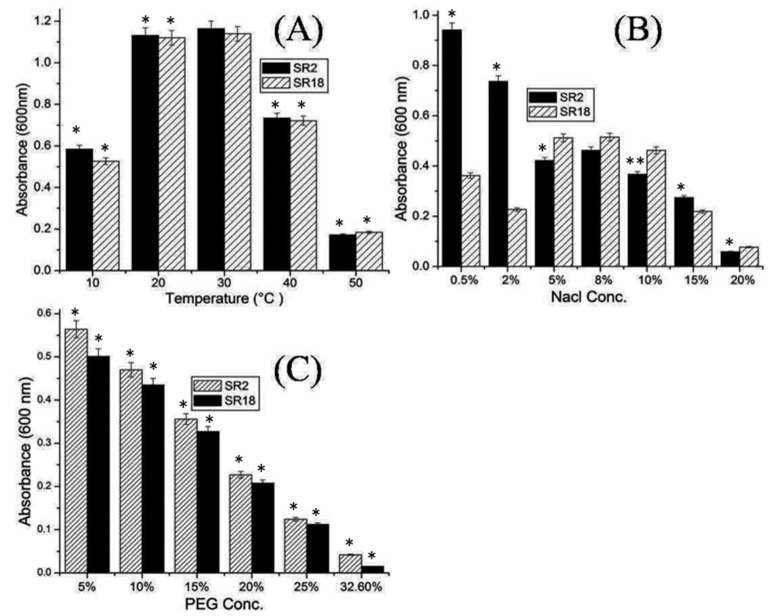
Abiotic stress tolerance activities of potential *Bacillus* strains SR2 (*Bacillus licheniformis*) and SR18 (*Bacillus velezensis*) (**A**) Temperature (10 to 50 °C) (**B**) Salinity Nacl (0.5 to 20%) (**C**) Drought PEG (5 to 32.6%). Experimental data are averages of three replicates. Bars display mean ± SE and statistical analysis was performed using a one-way ANOVA with a Tukey post hoc test. Differences between means were considered to be significant if the calculated *p*-values were *p* < 0.05 ** and *p* < 0.01 *.

**Figure 3 microorganisms-11-01085-f003:**
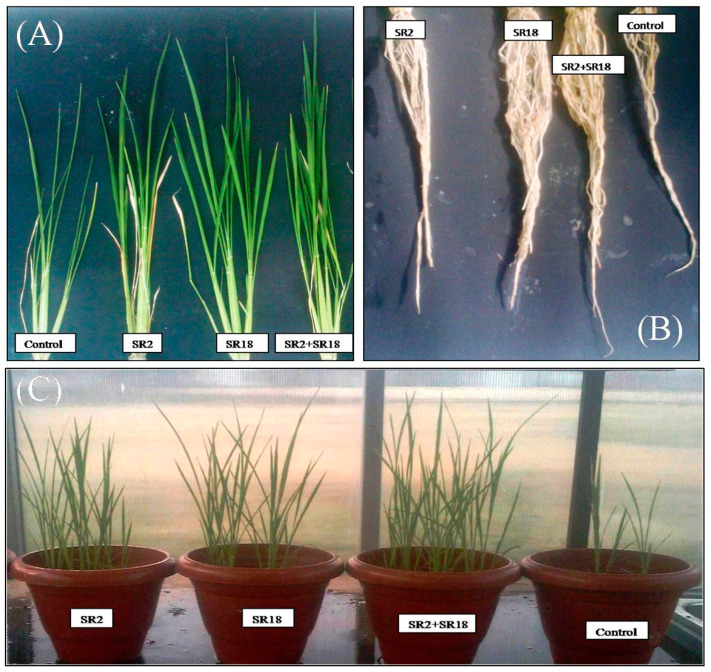
Effect of single inoculation and coinoculation of *Bacillus* strains on different growth parameters of rice plants (**A**) Comparison of shoot length of rice plants: control, SR2, SR18, SR2, and SR18 (**B**) Comparison of root length of rice plants (**C**) Comparison of overall plant growth promotion of rice plants.

**Table 1 microorganisms-11-01085-t001:** Multifarious PGP traits of potential *Bacillus* spp. isolated from soils of the IGP region.

Isolate	IAA (µg/mL)	PSI (mm)	Siderophore ^a^	Biocontrol		HCN ^b^	Ammonia ^b^	ACC ^c^	Cellulase	Protease	Lipase	Amylase
				*R. Solani*	*F. Oxysporum*							
SR2	90.64	11	++	62%	67%	-	+	++	Positive	Positive	Positive	Positive
SR3	56.32	2	++	-	29%	+	+	+	Positive	-	-	Positive
SR11	67.56	5	++	-	-	-	-	-	-	-	-	-
SR13	23.67	2	+	32%	38%	-	+	-	-	Positive	Positive	Positive
SR18	87.15	8.5	++++	50%	48%	++	+	++	-	Positive	Positive	Positive
SR20	40.30	5	+	-	-	-	+	+	-	-	Positive	-
SR56	17.12	2	++	31%	35%	-	-	+	Positive	-	-	Positive
SR58	25.43	-	+	-	-	-	+	-	-	positive	-	-

IAA: Indole-3-acetic acid, HCN: Hydrogen cyanide, ACC: 1-Aminocyclopropane-1-carboxylic acid. Each test listed above is performed in triplicates. a (-: negative, +: positive (hazy zone), ++: clear halo <5 mm, ++++: large halo >10 mm). b (-: negative, +:weak producer, ++:strong producer). c (-: negative, +:slow growth, ++:rapid growth).

**Table 2 microorganisms-11-01085-t002:** Morphological and biochemical characteristics of potential *Bacillus* strains SR2 and SR18 recovered from soils of IGP region.

S.No.	Property	Bacillus Strains	
		SR2	SR18
1	Gram nature	+, rods	+, rods
2	Endospore	Present	Present
3	Colony morphology	Colonies are irregular form, lobate margin, with bulging droplets of mucilaginous growth.	Colonies are larger, undulate margin, with circular form and flat elevation.
4	Motility	+ve	+ve
5	Oxidase	+ve	−ve
6	Catalase	+ve	+ve
7	Indole	−ve	−ve
8	Methyl red	+ve	−ve
9	Voges-Proskauer	+ve	+ve
10	Citrate	+ve	−ve
11	Starch hydrolysis	+ve	+ve
12	Gelatin hydrolysis	+ve	+ve
13	Urease	−ve	−ve
14	Nitrate	+ve	+ve

**Table 3 microorganisms-11-01085-t003:** Closest sequences, % similarity and accession no. of the potential *Bacillus* strains by 16 s rRNA gene analysis.

S.No.	Bacillus Strains	Closest Sequence	% Similarity	*e*-Value	Genebank Accession No.
1.	SR2	* Bacillus licheniformis *	99%	0.00	KM052376
2.	SR18	*Bacillus velezensis*	99.51%	0.00	KM052377

**Table 4 microorganisms-11-01085-t004:** Effect of single inoculation and coinoculation of *Bacillus* strains on different growth parameters of rice plants under greenhouse conditions.

Treatment	Growth Parameters							
	Shoot Length (cm)	Shoot Dry Weight (g/plant)	Tiller Number	Leaf Number	Root Length (cm)	Root Number	Root Dry Weight (g/plant)	Yield (g/plant)
Control (Untreated)	45.0 ± 1.0 ^a^	0.288 ± 0.02 ^a^	3.3 ± 0.5 ^a^	12 ± 1.0 ^a^	17.8 ± 1.15 ^a^	51.6± 7.6 ^a^	0.170 ± 0.02 ^a^	24.40 ± 2.10 ^a^
A (SR2)	51.6 ± 0.5 ^b^	0.441 ± 0.01 ^ab^	4.6 ± 0.5 ^ab^	16.6 ± 2.08 ^b^	23.3 ± 0.76 ^bc^	69.6 ± 8.5 ^b^	0.231 ± 0.02 ^ab^	26.33 ± 0.76 ^a^
B (SR18)	52.7 ± 2.5 ^b^	0.543 ± 0.00 ^b^	7.0 ± 1.0 ^ab^	21.3 ± 1.52 ^c^	21.4 ± 0.55 ^ab^	161.3 ± 3.5 ^c^	0.313 ± 0.00 ^b^	27.53 ± 2.17 ^ab^
A + B (SR2 + SR18)	57.2 ± 1.16 ^c^	0.548 ± 0.01 ^b^	8.3 ± 0.5 ^b^	25.3 ± 2.08 ^d^	27.1 ± 1.86 ^c^	158.3 ± 7.6 ^c^	0.326 ± 0.15 ^b^	29.17 ± 3.27 ^b^

Experimental data are the average of three replicates. Means with different letters in the same column differ significantly at *p* < 0.05 according to Duncan’s multiple range test.

## Data Availability

Not applicable.

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
