# Peer review of "Screening for Multifarious Plant Growth Promoting and Biocontrol Attributes in Bacillus Strains Isolated from Indo Gangetic Soil for Enhancing Growth of Rice Crops"

_microorganisms, 2023, doi:10.3390/microorganisms11041085_

Round 1
Reviewer 1 Report (Previous Reviewer 2)
The authors responded to most of the comments and the quality of the manuscript has been improved.
However, there are still comments on the manuscript.
How were the seeds inoculated with a bacterial suspension of two strains?
How was the root length measured?
Line 262-263. “Various parameters viz., shoot length, root length, dry weight, and yield were studied during the course of the experiment”. Why are there no yield results?
The results obtained by the authors were described, but hardly discussed. This is a significant drawback. For example, why was strain SR18 less resistant to 0.5% and 2% NaCl than 5%, 8%, and 10% NaCl? Why did the authors not discuss which strain was more effective, nor the results of co-inoculation of seeds with both strains?
Author Response
Response to Reviewer 1 Comments
We have modified the manuscript accordingly and detailed corrections are listed below point by point:
Comments/points:
Point 1: How were the seeds inoculated with a bacterial suspension of two strains?
Response 1: For this purpose, an equal volume of two strains was mixed instantaneously before use. The resulting suspension was then mixed with 1% CMC to form a slurry which was coated on to the surface of the rice seeds.
Information in this regard has been incorporated in para 2.8.2 of material and method section of the manuscript
Point 2: How was the root length measured?
Response 2: Root length was measured using a meter scale. To determine root length, three individual plants were randomly chosen from each experimental pot and measured with a meter scale after uprooting the selected plants.
Relevant information in this regard has been included in para 2.8.1. of the material and method section of the manuscript.
Point 3: Line 262-263. “Various parameters viz., shoot length, root length, dry weight, and yield were studied during the course of the experiment”. Why are there no yield results?
Response 3: Thank you very much for your valuable comment. Regarding this, the required information has been incorporated in para 3.6 of the result and discussion section of the manuscript and Table No. 4 (last column).
Point 4: The results obtained by the authors were described, but hardly discussed. This is a significant drawback. For example, why was strain SR18 less resistant to 0.5% and 2% NaCl than 5%, 8%, and 10% NaCl?
Response 4: First of all, thank you for pointing out the significant drawback in the manuscript. As per your suggestion, information in this regard has been incorporated in the result and discussion section of the manuscript.
Further, it is also pertinent to mention that in the present study, the SR18 strain has been identified as Bacillus velezensis and its less resistance (0.5% and 2% NaCl than 5%, 8%, and 10% NaCl), may be due to its ability to grow well in high salt concentrations. Our study also supports the results of various other researchers who have shown that Bacillus velezensis tolerates high salt concentration i.e. up to 11% NaCl (Mahdi et al., 2022).
Point 5: Why did the authors not discuss which strain was more effective, nor the results of co-inoculation of seeds with both strains?
Response 5: Thank you very much for your valuable comments. The relevant results and discussion have been incorporated in the para 3.6 of the result and discussion section of the manuscript.

Reviewer 2 Report (New Reviewer)
This is an interesting manuscript about the identification of Bacillus strains from the rhizospheric soils of Oryza sativa in Indo-Gangetic plains (IGPs) and the use of these strains for biocontrol activities in vitro and in vivo as well as plant growth promoting.
The present work was organized logically, and the results obtained were reliable and persuasive. However, I have some points that need to be addressed as follows.
1. Line 34: Please do not use abbreviations without an explanation at the first mention.
2. It would be better to add these subtitles "2.3.3. HCN production, 2.3.4. Ammonia production, and 2.3.5. ACC deaminase activity under the title "2.2. Characterization of soil bacterial isolates for various PGP attributes".
3. It is best to make the subtitle "2.4.4. Morphological and biochemical characterization" a major title in the manuscript.
4. Also, the subtitle "2.4.5. Molecular characterization of potential isolates".
5. Most of the obtained results need more deep discussion such as In vitro antifungal activity of potential Bacillus isolates, Abiotic stress tolerance activity of potential Bacillus strains recovered from soils of IGP region, and Effect of rice associated Bacilli on growth of rice plants growing under greenhouse conditions.
6. Please add the statistical analysis to the results in Figure 2.
7. The English language needs to be revised throughout the manuscript.
Author Response
Response to Reviewer 2 Comments
We have modified the manuscript accordingly and detailed corrections are listed below point by point:
Comments/points:
Point 1: Line 34: Please do not use abbreviations without an explanation at the first mention.
Response 1: Thank you very much for pointing out this error. Necessary changes have been made in the manuscript accordingly.
Point 2. It would be better to add these subtitles "2.3.3. HCN production, 2.3.4. Ammonia production, and 2.3.5. ACC deaminase activity under the title "2.2. Characterization of soil bacterial isolates for various PGP attributes".
Response 2: Regards for the valuable comments. The above-mentioned changes have been incorporated.
Point 3. It is best to make the subtitle "2.4.4. Morphological and biochemical characterization" a major title in the manuscript.
Response 3: Corrections have been made as per suggestion.
Point 4. Also, the subtitle "2.4.5. Molecular characterization of potential isolates".
Response 4: Corrections have been made as per suggestion.
Point 5. Most of the obtained results need more deep discussion such as In vitro antifungal activity of potential Bacillus isolates, Abiotic stress tolerance activity of potential Bacillus strains recovered from soils of IGP region, and Effect of rice associated Bacilli on growth of rice plants growing under greenhouse conditions.
Response 5: The manuscript has been updated accordingly and the information has been incorporated in the para 3.2, 3.5 and 3.6 of the result and discussion section.
Point 6. Please add the statistical analysis to the results in Figure 2.
Response 6: As per the suggestion, the statistical analysis has now been incorporated in the Figure no. 2 of the manuscript (The data obtained after experimentation was statistically evaluated using ANOVA with a Tukey post hoc test).
Point 7. The English language needs to be revised throughout the manuscript
Response 7: We have revised the whole manuscript carefully and tried to avoid any grammar or syntax error. In addition, we have asked several colleagues who are skilled authors of English language papers to check the English. We believe that the language is now acceptable for the review process.

Round 2
Reviewer 2 Report (New Reviewer)
Thanks for considering the suggestions from the previous version. The manuscript is acceptable from my point of view. I recommend accepting it for publication.
This manuscript is a resubmission of an earlier submission. The following is a list of the peer review reports and author responses from that submission.
Round 1
Reviewer 1 Report
Manuscript "Screening for Multifarious Plant Growth Promoting and Biocontrol Attributes in Bacillus strains isolated from Indo Gangetic Soil for Enhancing Growth of Rice Crops" by Devi et al. describes research aimed at developing a biological method of biostimulation and biocontrol of rice by constructing preparations based on strains of the genus Bacillus with potential growth promotion properties, the supply of nutrients and in vitro inhibition of the growth of phytopathogenic fungi of the species Rhizoctonia solani and Fusariun oxysporum.
The paper should include much more extensive discussion of extremophilic microorganisms adapted to conditions of extremely high osmotic pressure and temperature and extremely low pH values, respectively: halophiles, thermophiles and acidophiles. It is necessary to clarify what characteristics are responsible for these adaptations to these conditions.
L 40-41 – it is not an unambiguous matter that IAA has a positive effect on plant growth in low concentrations and in high concentrations may be harmful
L 43 – these compounds are produced not only by bacteria, but also by other microorganisms, mainly fungi included in the PGPF group
L82 do not use the expression "bacilli" but bacteria of/from the genus Bacillus
L 106 Ratio R:S is CFU of rhizosphere bacteria to CFU of bulk soil – change non rhizosphere on bulk soil
L 113 the composition of the medium, in particular the pH values, must be reported for the substrates for determining the dissolution capacity of phosphates.
L 218 what were the conditions for soil sterilization – how many times this process was carried out
L 234 – the description of the pot experiment is incomplete. Incubation time, humidity are not given. How is the dry matter of plant seedlings determined?
In the places of Sidhvan, Barkaccha, a positive rhizosphere effect was not achieved, because the value of the R:S ratio was about 1.0, and in the place of Bijarkal this value was small. What is the reason for the lack of influence of root secretions on the CFU abundance of microorganisms?
L 187-L191, Fig.2. The authors should present the abundance of tested microorganisms as CFU/mL obtained as a result of individual cultures in different conditions of temperatures, salinity and water potentials on a solid medium
Figs. 3. Y axes are badly described. Growth promotion is a conclusion, not a description. On the Y axis of diagram A should be the length of the roots and leaves, and the graph B- dry weight of roots and leaves
Reviewer 2 Report
The manuscript is devoted to the topic of current interest of searching for, studying the properties and the possibility of using PGPR to increase crop yields. However, to improve its quality, I propose to make the following corrections to it.
- It is necessary to decipher the abbreviations (CAS, DF, IMViC, etc.). In addition, there are unnecessary abbreviations in the text that should be deleted (eg MOP, DAP (line 415), P (line 278)).
- In the Material and methods section, you should specify the following:
- manufacturers of used reagents and ready-made nutrient media;
- composition of all culture media used;
- what universal primers were used, what method was used to isolate DNA, PCR conditions, what model was used when constructing the phylogenetic tree. It is necessary to provide references to the methods used in section 2.4.5;
- what is meant by processing A and B and what serves as a control (line 232);
- a specific cultivation time, not "overnight" (line 150);
- how long did the growing experience last;
- how watering was done during the experiment.
- More notes on the Material and methods section:
- sections 2.5.1. and 2.5.3 can be merged;
- in section 2.1.1 it is not necessary to specify the formula for calculating CFU;
- section 2.1.2 can be deleted because it is not clear why the authors isolated bacteria from bulk soil and determined the R:S ratio, if goal of the study was to isolate and characterize bacteria from rice rhizosphere soil (lines 81–84). Besides,readers are already well aware that the number of microorganisms in the rhizosphere is higher than in bulk soil;
- it is not clear from which soil (rhizosphere or bulk soil) 28 isolates were isolated and according to what specific criteria they were selected. The authors indicate that the selection was made according to morphological and gram staining features, but this is not enough to clearly understand why these 28 isolates were selected;
- Why do authors constantly use the term "potential" in relation to Bacillus strains? What do they mean? (lines 159, 187-188, 294, 298, etc.)?
- Why do the authors draw readers' attention (table 1) to the number of gram-negative rods and gram-positive cocci, if their goal was to isolate strains of bacilli that belong to gram-positive rods? Given this remark, as well as the above doubts about the need to determine R:S, I propose to delete table 1, and indicate the number of isolates in the text.
- Notes to table 2:
- the sequence of columns in the table does not match the sequence of presentation of the results in the text of the manuscript;
- indicate the units of measurement of the number of IAA once in the header of the table;
- the ability to solubilize phosphates is indicated as specific Phosphate solubilization index values ​​(as written in section 2.2.1), and not simply as the presence / absence of the ability.
- Table 4. The level of similarity of the nucleotide sequence of the 16S rRNA gene (98%) is insufficient to classify the SR18 strain as Bacillus subtilis (Kim M, Oh HS, Park SC, Chun J. Towards a taxonomic coherence between average nucleotide identity and 16S rRNA gene sequence similarity for species demarcation of prokaryotes, Int J Syst Evol Microbiol 2014; 64:346–35). It can only be said with certainty that strain SR18 belongs to the genus Bacillus. This can be seen even in the phylogenetic tree (Fig. 1), where the SR18 strain is located separately and does not form a common cluster with Bacillus subtilis strains.
- Figure 1 does not indicate the statistical significance of the branching order (in %). Also, on the dendrogram, it should be indicated which strains are typical and which are not.
- In the caption to Fig. 2, the letter designations of the drawings (A, B, C) are incorrectly indicated.
- In the caption to Fig. 2 there is no explanation of how the data are presented on the graphs. Are these averages ± SE or is it not?
- The description of the experimental results in Section 3.6 does not match the data in Fig.3. The figure shows data on leaves and roots, and in the text - on shoots and roots. I propose to remove Fig. 3 altogether, because the discussed data are presented in Table 5.
- The manuscript does not contain sections Discussion and Conclusion, which must be included in it.